# Rethinking Multidimensional Discriminator Output for Generative Adversarial Networks

## Abstract

The study of multidimensional discriminator (critic) output for Generative Adversarial Networks has been underexplored in the literature. In this paper, we generalize the Wasserstein GAN framework to take advantage of multidimensional critic output and explore its properties. We also introduce a square-root velocity transformation (SRVT) block which favors training in the multidimensional setting. Proofs of properties are based on our proposed maximal $p$-centrality discrepancy, which is bounded above by $p$-Wasserstein distance and fits the Wasserstein GAN framework with multidimensional critic output $n$. Especially when $n = 1$ and $p = 1$, the proposed discrepancy equals 1-Wasserstein distance. Theoretical analysis and empirical evidence show that high-dimensional critic output has its advantage on distinguishing real and fake distributions, and benefits faster convergence and diversity of results.

## 1 Introduction

Generative Adversarial Networks (GAN) have led to numerous success stories in various tasks in recent years Yang et al. (2022); Yu et al. (2022); Niemeyer & Geiger (2021); Chan et al. (2021); Han et al. (2021); Karras et al. (2020a); Nauata et al. (2020); Heim (2019). The goal in a GAN framework is to learn a distribution (and generate fake data) that is as close to real data distribution as possible. This is achieved by playing a two-player game, in which a generator and a discriminator compete with each other and try to reach a Nash equilibrium Goodfellow et al. (2014). Arjovsky et al. Arjovsky & Bottou (2017); Arjovsky et al. (2017) pointed out the shortcomings of using Jensen-Shannon Divergence in formulating the objective function, and proposed using the 1-Wasserstein distance instead. Numerous promising frameworks Li et al. (2017); Mroueh et al. (2017b); Mroueh & Sercu (2017); Mroueh et al. (2017a); Wu et al. (2019); Deshpande et al. (2019); Ansari et al. (2020) based on other discrepancies were developed afterwards. Although some of these works use critic output dimension $n = 1$, empirical evidence can be found that using multiple dimension $n$ could be advantageous. For examples, in Li et al. (2017) authors pick different $n$s $(16, 64, 128)$ for different datasets; In Sphere GAN Park & Kwon (2019) their ablation study shows the best performance with $n = 1024$. However, the reason for this phenomenon has not been well explored yet.

One contribution of this paper is to explore the properties of multidimensional critic output in the generalized WGAN framework. Particularly, we propose a new metric on the space of probability distributions, called *maximal $p$-centrality discrepancy*. This metric is closely related to $p$-Wasserstein distance (Theorem 3.9) and can serve as an alternative of WGAN objective especially when the discriminator has multidimensional output. In this revised WGAN framework we show that using high-dimensional critic output could make discriminator more informative on distinguishing real and fake distributions (Proposition 3.11). In classical WGAN with only one critic output, the discriminator push-forwards (or projects) real and fake distributions to 1-dimensional space, and then look at their maximal mean discrepancy. This 1-dimensional push-forward may hide significant differences of distributions in the shadow. Even though ideally there exists a "perfect" push-forward which reveals any tiny differences, practically the discriminator has difficulties to reach that global optimal push-forward Stanczuk et al. (2021). However, using $p$-centrality allows to push-forward distributions to higher dimensional space. Since even an average high-dimensional push-forward may reveal more differences than a

good 1-dimensional push-forward, this reduces the burden on discriminator. Specifically, we show that more faithful $p$-centrality functions returns larger discrepancies between probability distributions (Lemma 3.11).

Another novelty of this work is to break the symmetry structure of the discriminator network by compositing with an asymmetrical square-root velocity transformation (SRVT). In general architectures people assume that the output layer of discriminator is fully connected. This setup puts all output neurons in equal and symmetric positions. As a result, any permutation of the multidimensional output vector will leave the value of objective function unchanged. This permutation symmetry implies that the weights connected to output layer are somehow correlated and this would undermine the generalization power of the discriminator network Liang et al. (2019); Badrinarayanan et al. (2015). After adding the asymmetrical SRVT block, each output neuron would be structurally unique (Proposition 3.15). Our understanding is that the structural uniqueness of output neurons would imply their functionality uniqueness. This way, different output neurons are forced to reflect distinct features of input distribution. Hence SRVT serves as an magnifier which favors the use of high-dimensional critic output.

The novelty of this work is summarised as follows:

1. We propose maximal $p$-centrality discrepancy in a generalized WGAN formulation which facilitates the analysis of properties of multidimensional critic output. We have theoretically proved it as a valid metric to distinguish probability distributions and use it in GAN objectives;
2. We utilize an asymmetrical (square-root velocity) transformation to break the symmetric structure of the discriminator network, which empirically magnifies the advantage of high-dimensional critic output.
3. With the proposed discrepancy, we show that high-dimensional discriminator output can be advantageous on distinguishing real and fake distributions. It can potentially result in faster convergence and improve diversity of results;

## 2   Related work

**Wasserstein Distance and Other Discrepancies Used in GAN:** Arjovsky et al. Arjovsky et al. (2017) applied Kantorovich-Rubinstein duality for 1-Wasserstein distance as loss function in GAN objective. WGAN makes great progress toward stable training compared with previous GANs, and marks the start of using Wasserstein distance in GAN. However, sometimes it still may converge to sub-optimal optima or fail to converge due to the raw realization of Lipschitz condition by weight clipping. To resolve these issues, researchers proposed sophisticated waysGulrajani et al. (2017); Wei et al. (2018); Miyato et al. (2018) to enforce Lipschitz condition for stable training. Recently, people come up with another way to involve Wasserstein distance in GAN Wu et al. (2019); Kolouri et al. (2019); Deshpande et al. (2018); Lee et al. (2019). They use the Sliced Wasserstein Distance Rabin et al. (2011); Kolouri et al. (2016) to estimate the Wasserstein distance from samples based on a summation over the projections along random directions. Either of these methods rely on pushforwards of real and fake distributions through Lipschitz functions or projections on to 1-dimensional space. In our work, we attempt to distinguish two distributions by looking at their pushforwards in high dimensional space.

Another way people used to distinguish real data and fake data distributions in generative network is by moment matching Li et al. (2015); Dziugaite et al. (2015). Particularly, in Li et al. (2017) the authors used the kernel maximum mean discrepancy (MMD) in GAN objective, which aims to match infinite order of moments. In our work we propose to use the maximum discrepancy between $p$-centrality functions to measure the distance of two distributions. The $p$-centrality function (Definition 3.1) is exactly the $p$-th root of the $p$-th moment of a distribution. Hence, the maximal $p$-centrality discrepancy distance we propose can be viewed as an attempt to match the $p$-th moment for any given $p \geq 1$.

**$p$-Centrality Functions:** The mean or expectation of a distribution is a basic statistic. Particularly, in Euclidean spaces, it is well known that the mean realizes the unique minimizer of the so-called Fréchet function of order 2 (cf. Grove & Karcher (1973); Bhattacharya & Patrangenaru (2003); Arnaudon et al. (2013)). Generally speaking, a Fréchet function of order $p$ summarizes the $p$-th moment of a distribution with respect to any base point. A topological study of Fréchet functions is carried out in Hang et al. (2019)

which shows that by taking $p$-th root of a Fréchet function, the $p$-centrality function can derive topological summaries of a distribution which is robust with respect to $p$-Wasserstein distance. In our work, we propose using $p$-centrality functions to build a nice discrepancy distance between distributions, which would benefit from its close connection with $p$-Wasserstein distance.

**Asymmetrical Networks:** Symmetries occur frequently in deep neural networks. By symmetry we refer to certain group actions on the weight parameter space which keep the objective function invariant. These symmetries would cause redundancy in the weight space and affects the generalization capacity of network Liang et al. (2019); Badrinarayanan et al. (2015). There are two types of symmetry: (i) permutation invariant; (ii) rescaling invariant. A straight forward way to break symmetry is by random initialization (cf. Glorot & Bengio (2010); He et al. (2015)). Another way to break symmetry is via skip connections to add extra connections between nodes in different layers He et al. (2016a;b); Huang et al. (2017). In our work, we attempt to break the permutation symmetry of the output layer in the discriminator using a nonparametric asymmetrical transformation specified by square-root velocity function (SRVF) Srivastava et al. (2011); Srivastava & Klassen (2016). The simple transformation that converts functions into their SRVFs changes Fisher-Rao metric into the $L^2$ norm, enabling efficient analysis of high-dimensional data. Since the discretised formulation of SRVF is equivalent with an non-fully connected network, it can be viewed as breaking symmetry by deleting specific connections from the network.

## 3 Methodology

In this section we use the proposed GAN framework as a starting point to study the behaviors of multidimensional critic output.

### 3.1 Objective Function

The objective function of the proposed GAN is as follows:

$$\min_G \max_D \left(E_x[\|D(x)\|^p]\right)^{1/p} - \left(E_z[\|D(G(z))\|^p]\right)^{1/p} \tag{1}$$

where $\|\cdot\|$ denotes $L^2$ norm. $G$ and $D$ denotes generator and discriminator respectively. $p$ refers to the order of moments. $x \sim \mathbb{P}_r$ is the input real sample and $z \sim p(z)$ is a noise vector for the generated sample. The output of the last dense layer of discriminator is an $n$-dimensional vector in the Euclidean space $\mathbb{R}^n$. In contrast to traditional WGAN with 1-dimensional discriminator output, our framework allows the last dense layer of discriminator to have multidimensional output.

### 3.2 $p$-centrality function

The $p$-centrality function was introduced in Hang et al. (2019) which offers a way to obtain robust topological summaries of a probability distribution. In this section we show that $p$-centrality function is not only a robust but also a relatively faithful indicator of a probability distribution.

**Definition 3.1** ($p$-centrality function). Given a Borel probability measure $\mathbb{P}$ on a metric space $(M, d)$ and $p \geq 1$, the $p$-centrality function is defined as

$$\sigma_{\mathbb{P},p}(x) := \left(\int_M d^p(x,y)d\mathbb{P}(y)\right)^{\frac{1}{p}} = \left(\mathbb{E}_{y\sim\mathbb{P}}[d^p(x,y)]\right)^{\frac{1}{p}}.$$

Particularly, the value of $p$-centrality function at $x$ is the $p$-th root of the $p$-th moment of $\mathbb{P}$ with respect to $x$. As we know it, the $p$-th moments are important statistics of a probability distribution. After taking the $p$-th root, the $p$-centrality function retains those important information in $p$-th moments, and it also shows direct connection with the $p$-Wasserstein distance $W_p$:

**Lemma 3.2.** *For any $x \in M$, let $\delta_x$ be the Dirac measure centered at $x$. Then $\sigma_{\mathbb{P},p}(x) = W_p(\mathbb{P}, \delta_x)$.*

**Lemma 3.3.** *For any two Borel probability measures $\mathbb{P}$ and $\mathbb{Q}$ on $(M, d)$, we have*

$$\|\sigma_{\mathbb{P},p} - \sigma_{\mathbb{Q},p}\|_\infty \leq W_p(\mathbb{P}, \mathbb{Q}) \leq \|\sigma_{\mathbb{P},p} + \sigma_{\mathbb{Q},p}\|_\infty.$$

*Proof.* For any $x \in M$, by Lemma 3.2 and triangle inequality we have

$$|\sigma_{\mathbb{P},p}(x) - \sigma_{\mathbb{Q},p}(x)| \leq W_p(\mathbb{P}, \mathbb{Q}) \leq |\sigma_{\mathbb{P},p}(x) + \sigma_{\mathbb{Q},p}(x)|.$$

The result follows by letting $x$ run over all $M$. □

Let $\mathcal{P}(M)$ be the set of all probability measures on $M$ and let $C_0(M)$ be the set of all continuous functions on $M$. We define an operator $\Sigma_p : \mathcal{P}(M) \to C_0(M)$ with $\Sigma_p(\mathbb{P}) = \sigma_{\mathbb{P},p}$. Lemma 3.3 implies that $\Sigma_p$ is 1-Lipschitz.

Specifically, since $p$-Wasserstein distance $W_p$ metrizes weak convergence when $(M, d)$ is compact, we have:

**Proposition 3.4.** *If $(M, d)$ is compact and $\mathbb{P}$ weakly converges to $\mathbb{Q}$, then $\sigma_{\mathbb{P},p}$ converges to $\sigma_{\mathbb{Q},p}$ with respect to $L^\infty$ distance.*

*Remark* 3.5. On the other hand, if $\sigma_{\mathbb{P},p} \equiv \sigma_{\mathbb{Q},p}$, Lemma 3.2 implies $W_p(\mathbb{P}, \delta_x) = W_p(\mathbb{Q}, \delta_x)$ for any Dirac measure $\delta_x$. Intuitively this means that, at least, $\mathbb{P}$ and $\mathbb{Q}$ look the same from the point of view of all Dirac measures. This implies that $p$-centrality function is a relatively faithful indicator of a probability distribution.

### 3.3 The maximal $p$-centrality discrepancy

To measure the dissimilarity between two complicated distributions $\mathbb{P}$, $\mathbb{Q}$, we can consider how far the indicators of their push-forwards or projections $f_*\mathbb{P}$, $f_*\mathbb{Q}$ could fall apart. According to the dual formulation of $W_1$:

$$K \cdot W_1(\mathbb{P}, \mathbb{Q}) = \sup_{f \in Lip(K)} \mathbb{E}_{x \sim f_*\mathbb{P}}[x] - \mathbb{E}_{y \sim f_*\mathbb{Q}}[y],$$

even considering very simple indicators – the expectations – as long as we can search over all $K$-Lipschitz functions $f \in Lip(K)$, we can still approach $W_1$.

Even though a neural network is very powerful on generating all kinds of Lipschitz functions, it may not be able to or have difficulties to generate the optimal push-forward. This may affects the performance of WGANStanczuk et al. (2021). Hence if we consider more faithful indicators, is it possible to obtain more reliable fake distribution even using sub-optimal push-forward? Motivated by this, we consider Lipschitz functions $f : M \to \mathbb{R}^n$ and replace the expectations by the $p$-centrality functions. Particularly, for fixed base point $x_0 \in \mathbb{R}^n$ we look at discrepancy:

$$L_{p,n,K}(\mathbb{P}, \mathbb{Q}) := \sup_{f \in Lip(K)} \sigma_{f_*\mathbb{P},p}(x_0) - \sigma_{f_*\mathbb{Q},p}(x_0).$$

**Lemma 3.6.** *The definition of $L_{p,n,K}$ is independent of the choice of the base point. Or simply*

$$L_{p,n,K}(\mathbb{P}, \mathbb{Q}) = \sup_{f \in Lip(K)} \left( \int \|f\|^p d\mathbb{P} \right)^{\frac{1}{p}} - \left( \int \|f\|^p d\mathbb{Q} \right)^{\frac{1}{p}}.$$

*Proof.* Let $\phi$ be the translation map on $\mathbb{R}^n$ with $\phi(y) = y + x_0$. Then $g := \phi^{-1} \circ f \in Lip(K)$ iff. $f \in Lip(K)$ and

$$\begin{aligned}
L_{p,n,K}(\mathbb{P}, \mathbb{Q}) &= \sup_{f \in Lip(K)} \sigma_{f_*\mathbb{P},p}(\phi(0)) - \sigma_{f_*\mathbb{Q},p}(\phi(0)) \\
&= \sup_{f \in Lip(K)} \sigma_{(\phi^{-1} \circ f)_*\mathbb{P},p}(0) - \sigma_{(\phi^{-1} \circ f)_*\mathbb{Q},p}(0) \\
&= \sup_{g \in Lip(K)} \sigma_{g_*\mathbb{P},p}(0) - \sigma_{g_*\mathbb{Q},p}(0) \\
&= \sup_{f \in Lip(K)} \left( \int \|f\|^p d\mathbb{P} \right)^{\frac{1}{p}} - \left( \int \|f\|^p d\mathbb{Q} \right)^{\frac{1}{p}}.
\end{aligned}$$

□

The following proposition implies that $L_{n,p,K}$ is a direct generalization of Wasserstein distance:

**Proposition 3.7.** *If* supp$[\mathbb{P}]$ *and* supp$[\mathbb{Q}]$ *are both compact, then*

$$L_{1,1,K}(\mathbb{P}, \mathbb{Q}) = K \cdot W_1(\mathbb{P}, \mathbb{Q}).$$

*Proof.* Since $f \in Lip(K)$ implies $|f| \in Lip(K)$, we easily have $L_{1,1,K} \leq K \cdot W_1$.

On the other hand, for any $\epsilon > 0$, there exists a $K$-Lipschitz map $f : M \to \mathbb{R}$ s.t. $\int f d\mathbb{P} - \int f d\mathbb{Q} > K \cdot W_1(\mathbb{P}, \mathbb{Q}) - \epsilon$. Let $D = $ supp$[\mathbb{P}] \cup $ supp$[\mathbb{Q}]$ and $c = \min_{x \in D} f(x)$, then $f - c \geq 0$ and $\int f d\mathbb{P} - \int f d\mathbb{Q} = \int (f - c) d\mathbb{P} - \int (f - c) d\mathbb{Q} = \int |f - c| d\mathbb{P} - \int |f - c| d\mathbb{Q} \leq L_{1,1,K}(\mathbb{P}, \mathbb{Q})$. Hence $L_{1,1,K}(\mathbb{P}, \mathbb{Q}) \geq K \cdot W_1(\mathbb{P}, \mathbb{Q}) - \epsilon$ for any $\epsilon > 0$ which implies $L_{1,1,K} \geq K \cdot W_1$. $\qquad\square$

Recall that in WGAN, the discriminator is viewed as a $K$-Lipschitz function. In our understanding, this requirement is enforced to prevent the discriminator from distorting input distributions too much. More precisely, in the more general setting, the following is true:

**Proposition 3.8.** *Given any $K$-Lipschitz map $f : (M, d_M) \to (N, d_N)$ and Borel probability distributions $\mathbb{P}, \mathbb{Q} \in \mathcal{P}(M)$. Then the pushforward distributions $f_*\mathbb{P}, f_*\mathbb{Q} \in \mathcal{P}(N)$ satisfy*

$$W_p(f_*\mathbb{P}, f_*\mathbb{Q}) \leq K \cdot W_p(\mathbb{P}, \mathbb{Q}).$$

*Proof.* Let $\Gamma(\mathbb{P}, \mathbb{Q})$ be the set of all joint probability measures of $\mathbb{P}$ and $\mathbb{Q}$. For any $\gamma \in \Gamma(\mathbb{P}, \mathbb{Q})$, we have $f_*\gamma \in \Gamma(f_*\mathbb{P}, f_*\mathbb{Q})$. By definition of the $p$-Wasserstein distance,

$$\begin{aligned}
&W_p(f_*\mathbb{P}, f_*\mathbb{Q}) \\
&= \inf_{\gamma' \in \Gamma(f_*\mathbb{P}, f_*\mathbb{Q})} \left( \int_{N \times N} d_N^p(y_1, y_2) d\gamma'(y_1, y_2) \right)^{1/p} \\
&\leq \inf_{\gamma \in \Gamma(\mathbb{P}, \mathbb{Q})} \left( \int_{N \times N} d_N^p(y_1, y_2) d(f_*\gamma)(y_1, y_2) \right)^{1/p} \\
&= \inf_{\gamma \in \Gamma(\mathbb{P}, \mathbb{Q})} \left( \int_{M \times M} d_N^p(f(x_1), f(x_2)) d\gamma(x_1, x_2) \right)^{1/p} \\
&\leq \inf_{\gamma \in \Gamma(\mathbb{P}, \mathbb{Q})} \left( \int_{M \times M} K^p \cdot d_M^p(x_1, x_2) d\gamma(x_1, x_2) \right)^{1/p} \\
&= K \cdot \inf_{\gamma \in \Gamma(\mathbb{P}, \mathbb{Q})} \left( \int_{M \times M} d_M^p(y_1, y_2) d\gamma(y_1, y_2) \right)^{1/p} \\
&= K \cdot W_p(\mathbb{P}, \mathbb{Q}).
\end{aligned}$$

$\qquad\square$

More generally, $L_{n,p,K}$ is closely related with $p$-Wasserstein distance:

**Theorem 3.9.** *For any Borel distributions $\mathbb{P}, \mathbb{Q} \in \mathcal{P}(M)$,*

$$L_{p,n,K}(\mathbb{P}, \mathbb{Q}) \leq K \cdot W_p(\mathbb{P}, \mathbb{Q}).$$

*Proof.* By Lemma 3.2, we have

$$L_{p,n,K}(\mathbb{P}, \mathbb{Q}) = \sup_{f \in Lip(K)} W_p(f_*\mathbb{P}, \delta_0) - W_p(f_*\mathbb{Q}, \delta_0).$$

Applying triangle inequality and Proposition 3.8, we have

$$L_{p,n,K}(\mathbb{P}, \mathbb{Q}) \leq \sup_{f \in Lip(K)} W_p(f_*\mathbb{P}, f_*\mathbb{Q}) \leq K \cdot W_p(\mathbb{P}, \mathbb{Q}).$$

$\qquad\square$

Also $L_{n,p,K}$ is closely related with an $L^\infty$ distance:

**Proposition 3.10.** *For any $K$-Lipschitz map $f : M \to \mathbb{R}^n$,*

$$\|\sigma_{f_*\mathbb{P},p} - \sigma_{f_*\mathbb{Q},p}\|_\infty \leq \max\{L_{p,n,K}(\mathbb{P},\mathbb{Q}), L_{p,n,K}(\mathbb{Q},\mathbb{P})\}.$$

*Proof.*

$$\begin{aligned}
&\|\sigma_{f_*\mathbb{P},p} - \sigma_{f_*\mathbb{Q},p}\|_\infty \\
&= \sup_{x_0 \in \mathbb{R}^n} \left|\sigma_{f_*\mathbb{P},p}(x_0) - \sigma_{f_*\mathbb{Q},p}(x_0)\right| \\
&\leq \sup_{x_0 \in \mathbb{R}^n} \sup_{f \in Lip(K)} \left|\sigma_{f_*\mathbb{P},p}(x_0) - \sigma_{f_*\mathbb{Q},p}(x_0)\right| \\
&= \sup_{x_0 \in \mathbb{R}^n} \max\{L_{p,n,K}(\mathbb{P},\mathbb{Q}), L_{p,n,K}(\mathbb{Q},\mathbb{P})\}.
\end{aligned}$$

$\square$

The lower bound in Proposition 3.10 implies that, when we feed two distributions into the discriminator $f$, as long as some differences retained in the push-forwards $f_*\mathbb{P}$ and $f_*\mathbb{Q}$, they would be detected by $L_{p,n,K}$. The upper bound in Theorem 3.9 implies that, if $\mathbb{P}$ and $\mathbb{Q}$ only differ a little bit under distance $W_p$, then $L_{p,n,K}(\mathbb{P},\mathbb{Q})$ would not change too much.

As we increase $n$, the $p$-centrality function become more and more faithful which picks up more differences in the discrepancy:

**Proposition 3.11.** *If integers $n < n'$, then for any $\mathbb{P}, \mathbb{Q} \in \mathcal{P}(\mathbb{R}^m)$, we have $L_{p,n,K}(\mathbb{P},\mathbb{Q}) \leq L_{p,n',K}(\mathbb{P},\mathbb{Q})$.*

*Proof.* For any $n < n'$ we have natural embedding $\mathbb{R}^n \hookrightarrow \mathbb{R}^{n'}$. Hence any $K$-Lipschitz function with domain $\mathbb{R}^n$ can also be viewed as a $K$-Lipschitz function with domain $\mathbb{R}^{n'}$. Hence larger $n$ gives larger candidate pool for searching the maximal discrepancy and the result follows. $\square$

By Proposition 3.11 and Theorem 3.9, the limit

$$L_{p,K}(\mathbb{P},\mathbb{Q}) := \lim_{n \to \infty} L_{p,n,K}(\mathbb{P},\mathbb{Q})$$

exists and is bounded above by $K \cdot W_p(\mathbb{P},\mathbb{Q})$. Particularly, this bound is tight when $p = 1$ (Proposition 3.7).

As a summation, when we use weight regularization such that the discriminator is $K$-Lipschitz and fix some learning rate, using larger critic output dimension $n$ implies that:

1. the discriminator may get better approximation of either $L_{p,K}$ or $K \cdot W_p$;

2. the gradient descent may dive deeper due to larger discrepancy;

3. the generated fake distribution may be more reliable due to more faithful indicator.

*Remark* 3.12. Remember that our comparison is under fixed Lipschitz constant $K$. For example, we can easily scale up the objective function to obtain larger discrepancy, but it is not fair comparison anymore. Because when scaling up objective functions we in fact scaled up both the Lipschitz constant and the maximal possible discrepancy.

### 3.4 Square Root Velocity Transformation

Section 3.3 suggests us to consider high-dimensional discriminator output. However, if the last layer of discriminator is fully connected, then all output neurons are in symmetric positions and the loss function is permutation invariant. Thus the generalization power of discriminator only depends on the equivalence class obtained by identifying each output vector with its permutations Badrinarayanan et al. (2015); Liang et al.

(2019). Correspondingly the advantage of high-dimensional output vector would be significantly undermined. In order to further improve the performance of our proposed framework, we consider adding an SRVT block to the discriminator to break the symmetric structure. SRVT is usually used in shape analysis to define a distance between curves or functional data.

Particularly, we view the high-dimensional discriminator output $(x_1, x_2, \cdots, x_n)$ as an ordered sequence.

**Definition 3.13.** The signed square root function $Q : \mathbb{R} \to \mathbb{R}$ is given by $Q(x) = \text{sgn}(x)\sqrt{|x|}$.

Given any differentiable function $f : [0, 1] \to \mathbb{R}$, its SRVT is a function $q : [0, 1] \to \mathbb{R}$ with

$$q := Q \circ f' = \text{sgn}(f')\sqrt{|f'|}. \tag{2}$$

SRVT is invertible. Particularly, from $q$ we can recover $f$:

**Lemma 3.14.**

$$f(t) = f(0) + \int_0^t q(s)|q(s)|ds. \tag{3}$$

By assuming $x_0 = 0$, a discretized SRVT

$$S : (x_1, x_2, \cdots, x_n) \in \mathbb{R}^n \mapsto (y_1, \cdots, y_n) \in \mathbb{R}^n$$

is given by

$$y_i = \text{sgn}(x_i - x_{i-1})\sqrt{|x_i - x_{i-1}|}, i = 1, 2, 3, \cdots, n.$$

Similarly, $S^{-1} : \mathbb{R}^n \to \mathbb{R}^n$ is given by

$$x_i = \sum_{j=1}^i y_j |y_j|, i = 1, 2, 3, \cdots, n.$$

With this transformation, the pullback of $L^2$ norm gives

$$\|(x_1, \cdots, x_n)\|_Q = \sqrt{\sum_{i=1}^n |x_i - x_{i-1}|} \tag{4}$$

Applying SRVT on a high-dimensional vector results in an ordered sequence which captures the velocity difference at each consecutive position. The discretized SRVT can be represented as a neural network with activation function to be signed square root function $Q$ as depicted in Fig 1. Particularly, for the purpose of our paper, each output neuron of SRVT is structurally unique:

**Proposition 3.15.** *Any (directed graph) automorphism of the SRVT block leaves each output neuron fixed.*

*Proof.* View the SRVT block as a directed graph, then all output neurons has out-degree 0. By the definition of discritized SRVT, there is a unique output neuron $v_0$ with in-degree 1 and any two different output neurons have different distance to $v_0$. Since any automorphism of directed graph would preserve in-degrees, out-degrees and distance, it has to map each output neuron to itself. □

Also, the square-root operation has smoothing effect which forces the magnitudes of derivatives to be more concentrated. Thus, values at each output neuron would contribute more similarly to the overall resulting discrepancy. It reduces the risk of over-emphasizing features on certain dimensions and ignoring the rest ones.

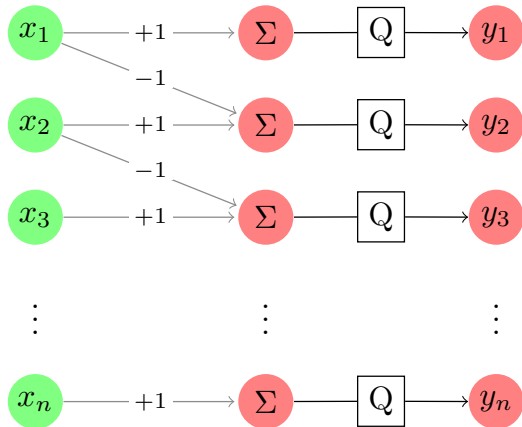

Figure 1: A representation of the SRVT block.

## 4 Experiments

In this section we provide experimental results supporting our theoretical analysis and explore various setups to study characteristics of multidimensional critic output. Final evaluation results on benchmark datasets are presented afterwards.

### 4.1 Implementation Details

We conducted image generation experiments with a number of settings. For unconditional generation task, we employed StyleGAN2 Karras et al. (2020b) and ResNetMiyato et al. (2018) architectures. In StyleGAN2 experiments we followed the default parameter settings provided by Karras et al. (2020b) except for $\gamma$ in $R_1$ regularization. In ResNet experiments we used spectral normalization to ensure Lipschitz condition. Adam optimizer was used with learning rate $1e - 4$, $\beta_1 = 0$ and $\beta_2 = 0.9$. The length of input noise vector $z$ was set to 128, and batch size was fixed to 64. For conditional generation task, we adopted BigGAN Brock et al. (2019) and used their default parameter settings. All training tasks were conducted on Tesla V100 GPUs.

### 4.2 Datasets and Evaluation Metrics

We implemented experiments on CIFAR-10, CIFAR-100 Krizhevsky et al. (2010), ImageNet-1K Deng et al. (2009), STL-10 Coates et al. (2011) and LSUN bedroom Yu et al. (2015) datasets. For each dataset, we center-cropped and resized the images, where images in STL-10, LSUN bedroom, and ImageNet were resized to $48 \times 48$, $64 \times 64$ and $256 \times 256$ respectively. Results were evaluated with Frechet Inception Distance (FID) Heusel et al. (2017), Kernel Inception Distance (KID) Bińkowski et al. (2018b) and Precision and Recall (PR) Sajjadi et al. (2018). Lower FID and KID scores and higher PR indicate better performance. In ablation study with ResNet architectures we generated 10K images for fast evaluation.In all other cases we used 50K generated samples against real sets for FID calculation. Precision and recall were calculated against test set for CIFAR-10 and validation set for ImageNet.

### 4.3 Results

In the following sections we first present ablation experimental results on CIFAR-10 with analysis, and then report final evaluation scores on all datasets.

**Ablation Study:**
We first studied the effect of multidimensional critic output using StyleGAN2 network architectures Karras et al. (2020b). Figure 2 shows recorded FID and $R_1$ penalty during training on CIFAR-10. Here we applied hinge loss as one common choice for settings with multidimensional output. From Figure 2 one can see higher $n$ led to faster convergence and consistently competitive results at all training stages. In training of

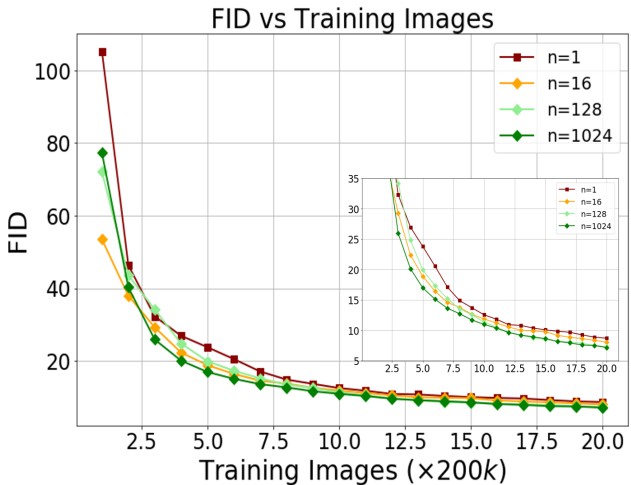

Figure 2: FID during training on CIFAR-10 with $n = 1, 16, 128$ and $1024$ using StyleGAN2 architectures.

StyleGAN2, $R_1$ regularization is used as a default choice for regularization. Note that successful training for higher $n$ in this case requires smaller $\gamma$s. In the experiments we used $\gamma = 1e-2, 1e-2, 1e-4, 1e-6$ for $n = 1, 16, 128, 1024$ respectively, where the total $R_1$ regularization term equals $0.5 \times \gamma \times R_1$ penalty. Figure 3 shows $R_1$ penalty during training under these settings.

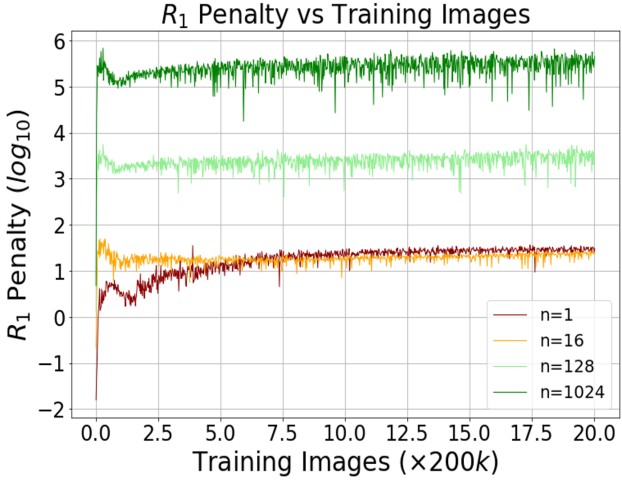

Figure 3: $R_1$ penalty during training on CIFAR-10 with $n = 1, 16, 128$ and $1024$ using StyleGAN2 architectures.

We then conducted experiments under different settings to explore the effects of $p$-centrality function and SRVT used in our framework. Since our approach is tightly related to WGAN, we also include results from WGAN-GP and WGAN-SN for comparison. In each setting we trained 100K generator iterations on CIFAR-10 using ResNet architectures, and reported average FID scores calculated from 5 runs in Fig 4. For this experiment we used 10K generated samples for fast evaluation. One can see without the use of SRVT (three green curves), settings with higher dimensional critic output resulted in better evaluation performances. The pattern is the same when comparing cases with SRVT (three blue curves). These observations are consistent with our Proposition 3.11. Furthermore, the results shows the asymmetric transformation boosts performances for different choices of $n$s, especially when $n = 1024$ (blue $vs$ green). Compared to WGAN where $n = 1$, for the same number of generator iterations, settings with multidimensional critic output produced images with better qualities, with nearly the same amount of training time. We observe generally a higher dimensional critic output $n$ requires less $n_{critic}$ to result in a stable training session in this case. This

is consistent with our theoretical results that a bigger $n$ leads to a "stronger" discriminator, and to result in a balanced game for the two networks, a smaller $n_{critic}$ can be used to benefit stable training sessions. The largest model (n = 1024) here results in 11 % more parameters ($\leq 2\%$ with n = 16 or 128) compared to WGAN-SN setting, which is in a reasonable range for comparison.

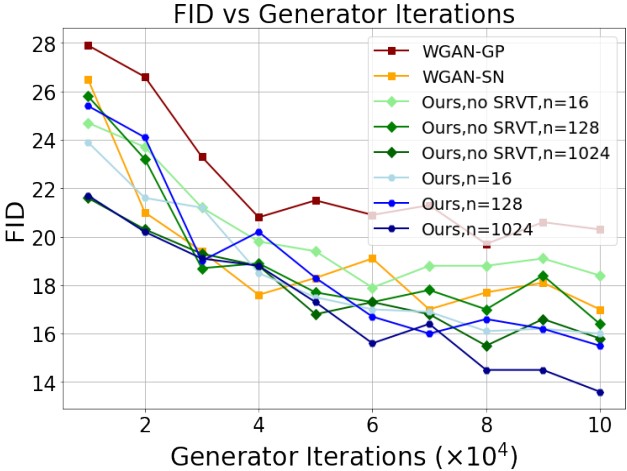

Figure 4: FID comparison under different settings during training using ResNet backbone.

In Fig 5 we present plots of precision and recall from these settings. For WGAN-GP we obtained $(0.850, 0.943)$ recall and precision. As we see the setting with the highest dimensional critic output $n = 1024$ and with the use of SRVT led to the best results compared to other settings. The result also indicates settings with high-dimensional critic output generated more diversified samples.

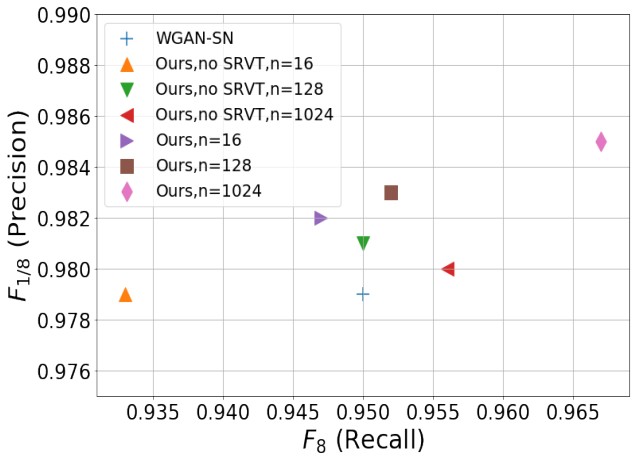

Figure 5: Precision and recall plot under different settings.

We also present comparisons using KID under different settings in Fig 6. Results in Fig 6(a) are aligned with previous evaluations which shows the advantage of using higher dimensional critic output. Performance was further boosted with SRVT. Fig 6(b) shows KID evaluations under different choices of $p$s, where SRVT was used with fixed $n = 1024$. We observe using $p = 1$ only, or both $p = 1$ and 2 resulted in better performance compared with using $p = 2$ only. In practice one can customize $p$ for usage. In the following we used $p = 1$ as the default setting. For the other two cases, we obtained (0.980,0.967) precision and recall for $p = 2$ only, and (0.987,0.965) when combining $p = 1$ and 2.

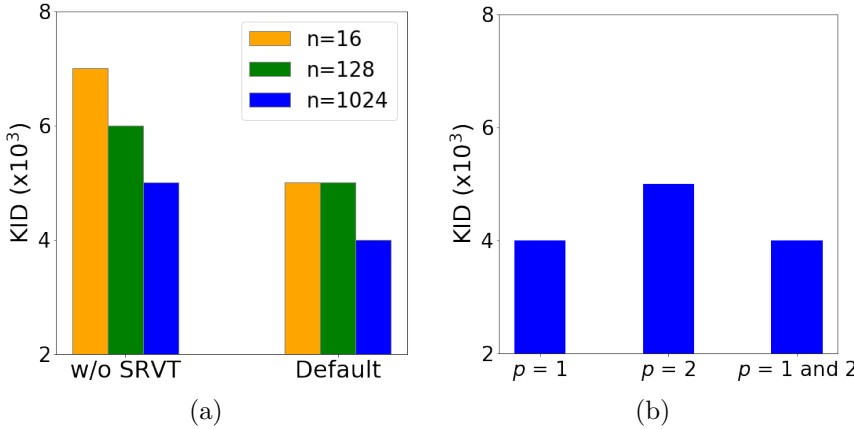

Figure 6: KID evaluation under different settings. (a) Left: without SRVT; Right: default setting with SRVT. (b) Evaluation with SRVT under different $p$s with fixed $n = 1024$.

We further conducted experiments to validate the effect of SRVT with MMD-GAN objective Li et al. (2017). For implementation we used the authors' default hyper-parameter settings and network architectures. From

Table 1: Evaluation of KID(x$10^3$)($\downarrow$) on the effect of SRVT with MMD-GAN objective and DCGAN architectures.

| Dimension of critic output $n$ | 16 | 128 | 1024 |
|---|---|---|---|
| w/o SRVT (Default) | 17(1) | 16(1) | 20(1) |
| w/ SRVT | 14(1) | 13(1) | 16(1) |

Table 1 one can see SRVT significantly boosts performance for different $n$s. The best result was obtained with $n = 128$ (default setup in [28]). We also notice for MMD-GAN, higher $n$ (1024) did not improve performance Bińkowski et al. (2018a), while we have shown our framework can take advantage of higher dimension critic output features.

In the following we display our final evaluation results. For fair comparison we list comparable results using the same network architectures.

**Quantitative Results:**
To compare GAN objectives, we present evaluations of FID on unconditional generation experiments averaged over 5 random runs in Table 2 . We compare with methods related to our work, including WGAN-GP Gulrajani et al. (2017), MMD GAN-rq Li et al. (2017), SNGAN Miyato et al. (2018), CTGAN Wei et al. (2018), Sphere GAN Park & Kwon (2019), SWGAN Wu et al. (2019), CRGAN Zhang et al. (2020) and DGflow Ansari et al. (2021).

Table 2: FIDs($\downarrow$) from unconditional generation experiments on CIFAR-10 with ResNet architectures.

| Method | CIFAR-10 | STL-10 | LSUN |
|---|---|---|---|
| WGAN-GP | 19.0(0.8) | 55.1 | 26.9(1.1) |
| SNGAN | 14.1(0.6) | 40.1(0.5) | 31.3(2.1) |
| MMD GAN-rq | - | - | 32.0 |
| CTGAN | 17.6(0.7) | - | 19.5(1.2) |
| Sphere GAN | 17.1 | 31.4 | 16.9 |
| SWGAN | 17.0(1.0) | - | 14.9(1.0) |
| CRGAN | 14.6 | - | - |
| DGflow | 9.6(0.1) | - | - |
| Ours | 8.5(0.3) | 26.1(0.4) | 14.2(0.2) |

As presented in Table 2, the proposed method led to competitive results in comparable settings on the three datasets.

Here we also present evaluation results of unconditional experiments on ImageNet using StyleGAN2 architectures. Table 3 shows the feasibility of using high-dimensional critic output in large-scale settings. With comparable FIDs, the precision-recall scores indicate that high-dimensional critic output potentially improves diversity of results.

| $n$ | FID | Precision | Recall |
|---|---|---|---|
| 1 | 55.82 | 0.677 | 0.883 |
| 1024 | 53.66 | 0.637 | 0.901 |

Table 3: Evaluations on $256 \times 256$ ImageNet experiments with $n = 1$ and 1024 using StyleGAN2 architectures.

For conditional generation, we show evaluation results from the original BigGAN setting and the proposed objective in Table 4. The results indicate the proposed framework can also be applied in the more sophisticated training setting and obtain competitive performance.

Table 4: FIDs($\downarrow$) from conditional generation experiments with BigGAN architectures.

| Objective | CIFAR-10 | CIFAR-100 |
|---|---|---|
| Hinge | 9.7(0.1) | 13.6(0.1) |
| Ours | 8.9(0.1) | 12.3(0.1) |

## 5  Broader Impact

Up to today majority of applications with GANs are adopting early frameworks with single critic output, possibly because those frameworks are easy to implement and can achieve relatively good performance. On the other hand, the properties of multidimensional critic output in GANs have not been well explored. We believe this paper may provide helpful evidence and insights for researchers to rethink about the area and explore its usage in future applications.

## 6  Conclusion and Discussion

In this paper we have explored the properties of multiple critic outputs in GANs based on the proposed the *maximal p-centrality discrepancy.* We have further introduced an asymmetrical (square-root velocity) transformation added to discriminator to break the symmetric structure of its network output. The use of the nonparametric transformation takes advantage of multidimensional features and improves the generalization capability of the network. Note that although the properties are investigated in a WGAN framework, the discovery can also be extended to other frameworks which utilize min-max discrepancy as objectives.

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
