# OpenReview forum: "Rethinking Multidimensional Discriminator Output for Generative Adversarial Networks"
_TMLR — Withdrawn by Authors_

### Review · Reviewer_K8AG · 2022-06-08

**Summary Of Contributions:**

This paper proposes to use multidimensional discriminator for training GANs. They first propose a new measure of distance between probability distributions which generalizes the Wasserstein distance and takes advantage of multidimensional outputs, that they call the maximal p-centrality discrepancy. They provide several proposition and bounds about the proposed distance and show how it relates to the Wasserstein distance.

To break the symmetry of the fully connected output layer, they propose to use a new layer that they call the Square Root Velocity Transformation (SRVT) that breaks the symmetry and permutation invariance of the output layer.

Finally they conduct several experiments with different GAN models on different datasets, where they show the benefits of using multi-dimensional discriminator. They then compare the multidimensional discriminator with and without the SVRT layer, and show how the SVRT layer improves performance.

**Broader Impact Concerns:**

This work doesn't have any direct ethical implications and thus does not require a Broader Impact Statement.

**Requested Changes:**

Main comments:
- In the introduction you mention Li et al. as well as Park & Kwon, which also seem to use a multidimensional discriminator. Can you explain how this paper is different from previous work using multidimensional discriminator.
- The motivation for the SVRT is not very clear, can you clarify the argument about the symmetry of the output layer and how SVRT addresses that. Also it's not very clear why you chose this particular transformation. Providing more intuition on what this transformation does might be useful. If it indeed breaks the symmetry some experimental measure of this would be useful.
- In proposition 3.11, I believe it's worth reminding the particular case when $p=1$ and $\text{supp}[\mathbb{P}] \text{ and } \text{supp}[\mathbb{Q}]$ are compact : where $L_{1,n,K}(\mathbb{P}, \mathbb{Q}) = KW_1(\mathbb{P}, \mathbb{Q}) \quad \forall n$. This is interesting because in this case there seems to be no point in using a multidimensional discriminator. Thus in the experiment when p=1, it's not clear why there is a benefit in using a multidimensional discriminator.
- At the end of section 3.3, I find the bullet points a bit confusing. In particular you're talking about learning rate and gradient descent, while the theory never use any of those concepts. So not sure what you're trying to conclude here from the theory ?
- Can you explain how you chose $\gamma$ in the experiments, and how does it influence the performance for the different values of n ?
- In the experiments you mention that higher dimensional critic requires less $n_{critic}$. You should define what $n_{critic}$ is, and do you have a plot that illustrate your claim ?
- Could you add confidence intervals for all of your plots ?

Minor comment:
- In the experiments could you explain how you use the hinge loss with the multidimensional outputs.
- Can you clarify what's the $p$ you're using in the different experiments ?
- In Table 1, what is the number between parentheses, is it the standard deviation or standard error ? if so you should mention this somewhere ?
- There is several typos and there is a problem with the Arjovsky citation where the name appears twice. There is also a problem with some citations missing a space between the citation and the text.

**Strengths And Weaknesses:**

Strength:
- The paper is well organized and the ideas are clearly presented.
- I enjoyed the section about the p-centrality discrepancy and how it relates to the Wasserstein distance.
- The experimental section is quite complete, with quite a lot of experiments which seems to all show the benefits of using multidimensional discriminator.

Weakness:
- The related work section could be improved. In particular, the author should better contrast the proposed distance with other multidimensional discriminator GANs such as MMD-GANs.
- The motivation for the multidimensional discriminator is not entirely clear. Apart from the performance benefits that the experiments seems to indicate, it's not clear if there is any other benefits and why the multidimensional discriminator improves the performance.
- The motivation for the SVRT was not clear to me. I didn't understand the argument about the symmetry of the output layer. Also it's not clear at all to me why you chose this particular transformation.
- Most of the experimental results and plots lack confidence intervals. And the experiments are sometimes missing some description, for example you mention that you use the Hinge loss, but you never explained how to use the Hinge loss with the multidimensional discriminator. You also chose different values for $\gamma$ but you don't explain why and how you chose those values.

---

### Review · Reviewer_xeL1 · 2022-06-12

**Summary Of Contributions:**

This paper focuses on Wasserstein GANs (WGANs, Arjovski et al. 2017). In WGAN, the discriminator model of the standard GAN framework represents a function with respect to which the Wasserstein distance between the real and fake distribution is computed, called ``critic'' within this framework. While Park and Kwon (2019), reported that using a larger dimension for the output of the critic improves performances, in this work the authors aim to further explore and understand this observation.

As a first contribution, this paper proposes a metric that is related to the p-Wasserstein distance---called *maximal p-centrality discrepancy*---which builds on the p-centrality function proposed by Hang et al. (2019).

As a second contribution, the paper introduces a so-called *square-root velocity transformation* (SVRT) block---usually used in shape analysis---for the architecture of the critic that is designed to break the symmetrical structure of the critic.

**Broader Impact Concerns:**

n/a.

**Requested Changes:**

Please discuss the above comments in weaknesses.

Below are some additional minor comments.

Minor comments:
- the citations should be with parenthesis where appropriate
- be more specific about which part the reader should look into when you are citing a side-contribution or a comment/discussion of related works (e.g. those citations in the introduction)
- Sec 1: make some statements such as "people assume that the output layer..." more precise (e.g. "the output layer of common implementations is a fully connected layer..." )
- Use consistently critic for WGAN (e.g. in Sec. 1 critic and discriminator are interchangeably used)
- explicitly give standard WGAN equation for completeness, or alternatively write down the difference below Eq. 1


Some example typos:
- Sec. 1: in several places, there is a missing "the" in front of "discriminator"
- Sec. 1: an magnifier -> a magnifier
- Sec. 1: last character : -> .
- Sec. 2: an non-fully -> a non-fully
- Sec. 3.3: even using ->  even when using
- Sec. 3.3: propasition
- Sec. 3.4: rest ones -> remaining ones

**Strengths And Weaknesses:**

=== Strengths ===

Overall, the general topic of considering different distances for the GAN framework is rather significant in my opinion: often GAN is a "clean" setup to explore ideas but the general problem of estimating distances between probability distributions using neural nets is widely applicable and relevant in machine learning.
Also, this paper does a nice balance between theoretical insights and experiments.

Below I list my main concerns.


==== Weaknesses ===

**Motivation for higher dimensional critic output.**
After reading the paper, it is unclear to me the motivation to use a higher dimensional output for the critic.
Owing to the Kantorovich-Rubinstein duality principle, the critic is the supremum over 1-Lipschitz functions. It is unclear what would one gain by making the output of this function higher-dimensional. In the write-up, this is motivated in Sec. 1 by arguing that a "push-forward" to one dimension may "hide significant differences". However, since the supremum is obtained in practice by backprop, I do not understand how could this claim hold (as the update per unit is proportional to the error it contributed to the output).
I am unaware of such a claim for the classification context, where in case the task is binary, it implies that using higher dimensional output improves performances.
From the theoretical side, while Prop. 3.11. argues for fixed Lip. const. the proposed discrepancy is larger when the output dimension is larger, it does not (directly) address the above question on the motivation. That is, one can use a larger step size to archive the same effect on the optimization (and for a smaller model size, see comment below). Hence, in short, I do not understand what is a drawback the baseline distance has which the proposed one explicitly resolves.

**On the SVRT contribution.**
To motivate the SVRT block, the authors point out that a symmetrical structure of the architecture reduces the generalization and cite two works that focus on standard DNN training (not GANs).
In my opinion, the generalization for GANs is not well understood, and it is not obvious that such conclusions (of the cited works) for minimization extend to GANs as well.
If by the above is meant the generalization of the critic (given fixed generator)---since the SVRT block is for the critic---it is not clear to me that that brings better "generalization" for the generator which is the model we ultimately are interested in.

**Writing \& presentation.**
The writing (in my opinion) is in some parts informal, which causes ambiguities when reading.
Some examples include: "high-dimensional critic output could make discriminator more informative on distinguishing"--where it is unclear what informative means; "1-dimensional push-forward may hide significant differences of distributions in the shadow"--it is very unclear what the authors technically mean. I recommend the authors rephrase or formalize their argument here.
Similarly, in  Sec. 3.3. I do not understand what it means that the gradient descent may  "dive deeper", or a given distribution to be more "reliable".


**Experiments.**
For a fair comparison, it may be more appropriate to consider an equal total number of parameters, as for example in Fig. 2 it is unclear if the gain comes from increasing the model size. Moreover, one can always add another layer to the critic, so that the last hidden layer is in effect similar to if one would increase the output size.
Also, it would be helpful to elaborate on how the step size for the baseline was selected. While it is noted that for some cases as increasing the output size the step size had to be reduced, it is unclear if larger step-size were explored for the baseline relative to that of the proposed method (see comment above for motivation for such experiments).

---

### Review · Reviewer_21Hq · 2022-06-12

**Summary Of Contributions:**

This paper studies the effect using multi-output discriminators for training Wasserstein Generative Adversarial Networks (WGANs). The main theoretical result is some characterizations of a Wasserstein-like distance induced by the p-th moment of n-output Lipschitz functions. The main experimental result shows that using n-output discriminators and a “SRVT” transform results in some improvements in image generation quality of WGANs measured by standard quantitative metrics.

**Broader Impact Concerns:**

I don’t see broader impact or ethical concerns regarding this paper.

**Requested Changes:**

Please find my questions above regarding the suggested improvements to both the theoretical results and the experiments.


**Strengths And Weaknesses:**

Strengths:

* GANs are known to be difficult to train, and basic training strategies (such as architectures, optimization algorithms, regularizers) have been an important driving factor for the progresses in GANs research. This paper reports some empirical advantages of using multi-output discriminators, which should be of interest to the community.

Weaknesses:

I. In my opinion, the theoretical results in this paper are quite far from giving a meaningful justification of multidimensional discriminators, and do not provide new insights about WGANs.

* The main proposal within the theory is a new proposed distance $L_{p,n,K}$, the maximum difference from the $p$-th moment of a $n$-output $K$-Lipschitz discriminator. However as the authors mentioned, when $p=1$, this equals the usual 1-Wasserstein distance, and does not depend on $n$. This fact itself already renders the theory very questionable (or at least fragile)—then large output dimension $n$ is helpful only for $p>1$. For one thing, it is unclear why $p>1$ is necessary (over $p=1$) from the first place, at least from looking at this paper.

* The main theoretical result (on Page 6, after Proposition 3.11) is that $L_{p,n,K}$ monotonically approaches a limit $L_{p,K}$ as $n\to\infty$, and $L_{p,K}$ is upper bounded by $K$ times the $p$-Wasserstein distance.
The authors then use this to argue multi-output discriminator is helpful, which I think has too many non-rigorous conceptual leaps and is far from a convincing scientific argument to me. For example,
  * “Using larger critic output $n$ implies that 1. The discriminator may get better approximation of either $L_{p,K}$ or $K\cdot W_p$” What does it mean by “discriminator getting better approximation of a distance”? If I understand it as “neural net discriminator found by standard GAN training”, then why does the above limit theorem implies this claim, that larger $n$ is helpful?
  * “2. The gradient descent may dive deeper due to larger discrepancy”. What does “gradient descent dive deeper” mean?

* The proofs of theoretical results look like standard math exercises of manipulating the definitions, and do not seem to give new analyses insights. While this on its own is not necessarily concerning, combined with the above concerns about the interpretations, this makes me further question the value of the theory presented in this paper.

II. The experimental results are significantly lacking in basic descriptions of the proposed method (as compared with standard contemporary ML papers), and discussions of its relationship with existing work (even though some existing methods are compared). Without these I don’t think the reported experimental results could stand for the claim.

Some examples are:

* The experimental claim is that multi-output discriminators are helpful. However Section 4.1-4.2 does not say what loss function is used for training multi-output discriminators. Is it the $L_{p,n,K}$ proposed in Section 3?

* As mentioned in the paper, multi-output discriminators have been tried in existing work, such as Li et al. (2017). What is their method, and are the experiments in this paper different from theirs?

* What is the $R_1$ regularization (first appearance on Page 8) and what is the $\gamma$ in it?

* How are hyperparameters tuned for different output dimension $n$, other than the $\gamma$ mentioned on Page 9?

Overall, with the above concerns, I think the paper is not ready for publication at TMLR.

---

### Note · Authors · 2022-06-13

**Comment:**

We sincerely thank the editor and reviewers for their efforts spent on the paper. After carefully reading the reviews, we decide the withdraw the current submission.

**Withdrawal Confirmation:**

I have read and agree with the venue's withdrawal policy on behalf of myself and my co-authors.